# Periodic Disturbance Compensation Control of a Rope-Driven Lower Limb Rehabilitation Robot

**Zhijun Wang \*, Mengxiang Li**  **and Xiaotao Zhang**

College of Mechanical Engineering, North China University of Science and Technology, Tangshan 063210, China; mxli@stu.ncst.edu.cn (M.L.); xtzhang@stu.ncst.edu.cn (X.Z.)
\* Correspondence: zjwang@ncst.edu.cn

**Abstract:** In order to solve the external periodic disturbance and unknown dynamics influence in the passive rehabilitation process of a rope-driven lower limb rehabilitation robot, a control method with periodic repeated learning was designed. In this control method, the closed-loop dynamics are divided into a periodic disturbance term, an unknown dynamics term, and a basic term, and the shape function is designed by using the Stone–Weirstrass theorem. In the process of periodic operation, the estimated value of the shape function coefficient is repeatedly learned to obtain the periodic disturbance term approximation and to realize the compensation in advance. Through the design of the impedance learning rate, the unknown dynamic term is periodically learned, and the unknown dynamics approximation is obtained. By combining the two approximations with the basic terms which can be solved directly, the external periodic disturbance is compensated for in advance and the control precision is improved. The control algorithm was verified by simulation, and the error fluctuation of the system gradually decreases and reaches the ideal value within several cycles. The performance of the control system is stable, and the problem of limb impedance caused by different patients is well solved.

**Keywords:** lower limb rehabilitation robot; compensation for external periodic disturbances; repetitive learning control; trajectory planning; Stone–Weirstrass theorem



## 1. Introduction

At the 67th World Health Assembly, the draft of the WHO Global Disability Action Plan for 2014–2021: Improving the Health of All Persons with Disabilities shows that over 1 billion people worldwide suffer from some form of disability, of which 100 to 200 million have extremely severe functional impairments. With the increasingly serious aging of the global population and continuous regional wars, the number of people with physical disabilities will gradually increase. In addition to being discriminated against due to their own shortcomings, people with disabilities also lack health care and rehabilitation services. Therefore, the research and development of rehabilitation facilities is becoming increasingly important. In China, compared with various data from 2011 and 2019, the number of rehabilitation institutions increased by 74%, the total number of rehabilitation personnel increased by 179.90%, and the proportion of business personnel increased from 69.18% to 71.70%. The ratio of rehabilitation institution personnel to disabled persons has increased from 40.0 people per 10,000 people to 70.2 people per 10,000 people [1]. Under the dual influence of the large number of disabled people and the continuous improvement of medical standards, the demand for rehabilitation robots is increasing.

Traditional rehabilitation training such as manual massage and orthosis has high labor intensity, poor sustainability, and weak targeting, which requires a large amount of energy from medical personnel. Moreover, the rehabilitation process highly relies on the experience of doctors, which is far from meeting the current rehabilitation needs [2]. With the rapid development of artificial intelligence technology, control engineering, and

medical levels, lower limb rehabilitation robots have an intelligent feedback system and stable control system, which can provide high-intensity and repetitive targeted treatment, greatly relieving medical pressure, and they have been widely used as an auxiliary facility of rehabilitation medicine. The Chicago Rehabilitation Research Center has developed a gait training and balance training robot, KineAssist [3], as shown in Figure 1. The robot has a flexible support system that uses power to drive the position of the support arm to change the ground support reaction force and control pelvic spatial movement and horizontal rotation through pelvic restraint devices, and it adopts a mixed use of active and passive joints to avoid forced injuries during the rehabilitation process and can achieve gait simulation and load training modes. However, the overall structure of the robot is complex and expensive, and the flexibility of the active and passive joints is poor, resulting in less-than-ideal practical promotion. In the research on rehabilitation robot technology carried out by the Fraunhofer Institute in Germany, some research results have been achieved in the field of rope traction rehabilitation robot technology [4], as shown in Figure 2. The rehabilitation robot controls the trunk through rope drive, with four ropes pulling downwards and three ropes pulling upwards. During the driving process, the trunk motion parameters are measured, and then the rope tension is adjusted using the motion parameters, which have good flexibility, and weight reduction training and strength training can be realized during gait training. The robot has a simple structure, a wide range of motion space, and high flexibility. However, due to large external disturbance and the poor stability of the rope drive, the control accuracy is relatively low. Therefore, in order to meet the requirements of high flexibility and low cost for rehabilitation robots, research on the control accuracy of rope-driven lower limb rehabilitation robots is becoming increasingly important.

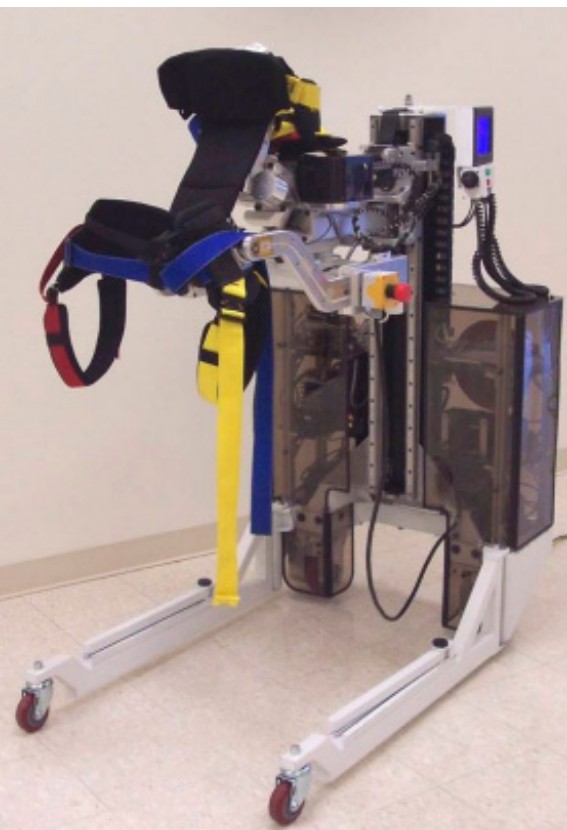

**Figure 1.** KineAssist rehabilitation robot.

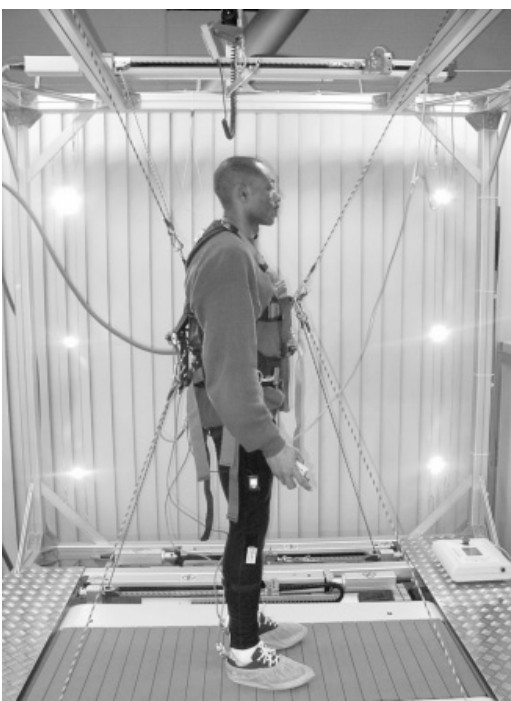

**Figure 2.** String-Man configuration.

Due to the differences in various physical parameters, degree of damage, and rehabilitation stages among different patients, multiple corresponding rehabilitation plans are required. Therefore, lower limb rehabilitation robots generally have problems of low control accuracy and poor universality [5,6]. In order to meet the requirements of adaptive control for lower limb rehabilitation robots in different patient rehabilitation processes, a periodic repetitive learning control was designed using external periodic cyclic disturbances [7] and the method of solving approximate values of unknown dynamics and periodic disturbances. Corresponding to the muscle tension, joint friction, and mechanical equipment errors that may be generated by different patients [8], advance compensation and fitting estimation values were carried out. The implementation of lower limb rehabilitation robots has high accuracy and universality characteristics.

The organization of this article is as follows. After the introduction, in Section 2, the structural design and some parameters of a rope-driven lower limb rehabilitation robot are introduced. In Section 3, we explain the problem of external periodic disturbances, collect human gait trajectory parameters, and design a control system. In Section 4, the stability of the repetitive learning control system was analyzed and proven. Presentation of simulation experiment results, in Section 5. The paper is concluded in Section 6, summarizing the present work.

## 2. Structural Design of the Lower Limb Rehabilitation Robot Driven by Rope

Compared with exoskeleton robots, rope-driven lower limb rehabilitation robots have the advantages of wide motion range, low inertia force, and high control accuracy [9]. The overall structure design uses an industrial aluminum profile, 5050L-8, to build the frame and adopts an S7-1200 PLC, a V90-PTI servo motor, and a 1204 screw drive to form the driving unit, as shown in Figure 3. The single leg is controlled by the first three and the last three, a total of six groups of driving units. The human body is suspended from the center of the frame by a five-point safety rope and stands on the walking machine. In terms of drive control, the C++ control algorithm is written, and the calculated results are used to communicate with PLC through Snap7 (an open-source software package based on the s7 communication protocol) [10,11] to transmit and receive the driving parameters and current motion parameters of the rehabilitation robot. After receiving the data, the PLC

changes the DI10 pin value of the V90-PTI servo motor to realize the switching of the torque and speed control mode and then complete the force control through the analog output. The rehabilitation process of the rope-driven lower limb rehabilitation robot is achieved by scaling four ropes on the thigh in the same plane to perform flexion/extension motion on the hip joint and by using ropes on two lower legs to achieve flexion/extension motion on the knee joint. This is used to simulate the gait posture of patients walking in a straight line, thereby achieving rehabilitation training for some motor functions. The main parameters of the lower limb rehabilitation robot driven by ropes are shown in Figure 4 and Table 1.

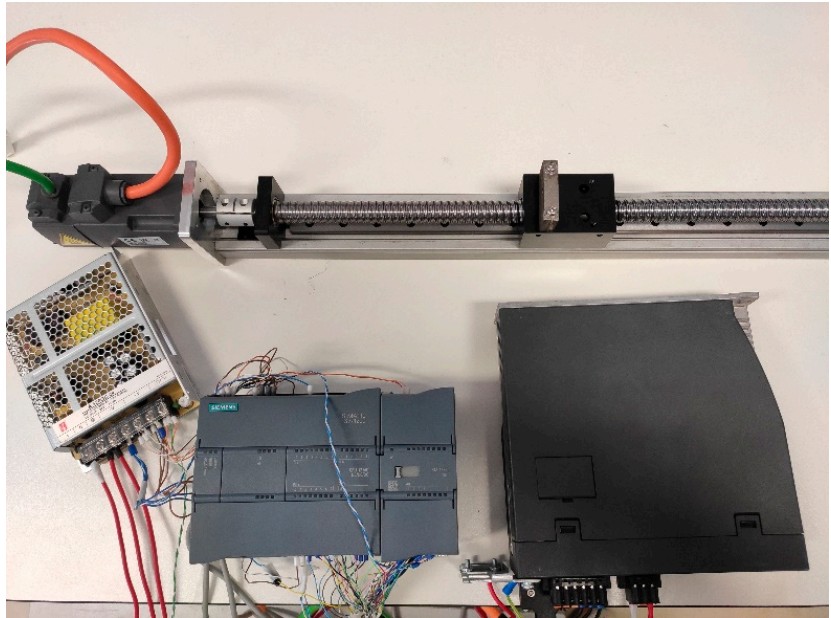

**Figure 3.** Drive unit.

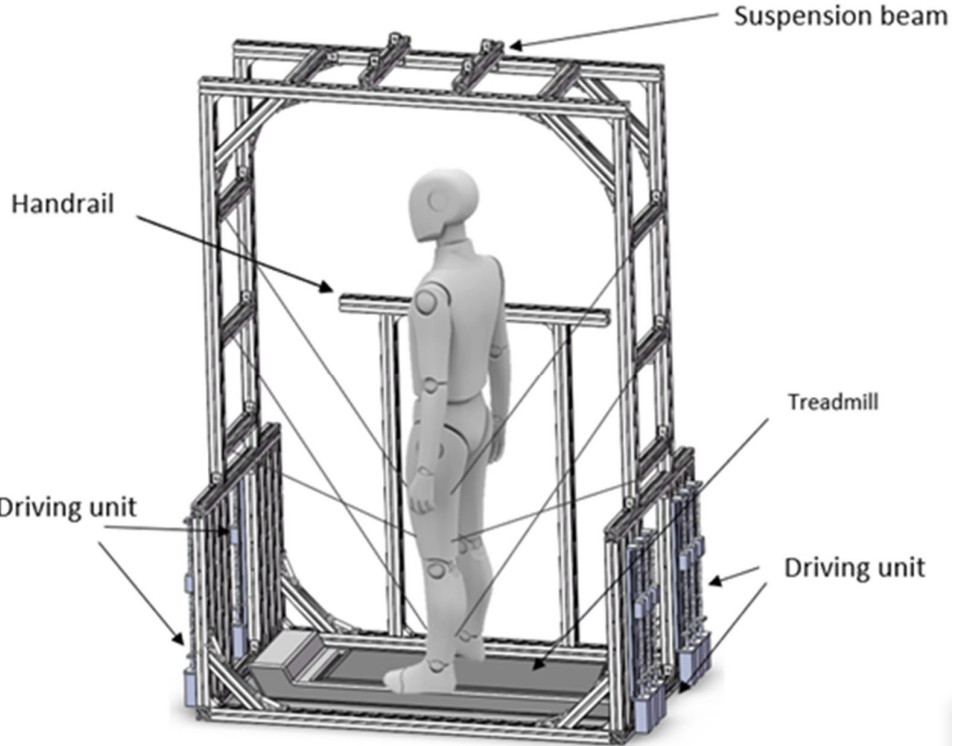

**Figure 4.** Rope-driven lower limb rehabilitation robot framework.

**Table 1.** Main parameters of the robot framework.

| No. | Item Parameter | Numerical Value |
|:---:|:---:|:---:|
| 1 | Frame size | 1600 mm × 1000 mm × 2350 mm |
| 2 | Activity space: | 1200 mm × 800 mm × 1900 mm |
| 3 | Maximum pulling force of the drive unit | 400 N |
| 4 | Suspension measures maximum load bearing | 480 kg |
| 5 | Applicable height range | 1.6~1.9 m |
| 6 | Applicable weight range | 35~80 kg |
| 7 | Material: alloy aluminum profile | 5050 L-8 |
| 8 | Total mass | 53.7 kg |

## 3. External Periodic Disturbance Control Design

### 3.1. Tracking Control Problem Description

Human lower limb rehabilitation training is generally divided into two parts. In the case of complete loss of motor function, external assistance is usually used to realize passive rehabilitation [12] of gait simulation. In this rehabilitation process, the joints carry out a repetitive round-trip motion, and the round-trip cycle is determined by the rehabilitation gait trajectory set in advance. For the control system, in the process of realizing the reciprocating motion, the periodic interference from the outside is the biggest uncertain factor. This unknown disturbance mainly comes from the supporting resistance of the sole of the foot, the tensioning force generated by the damaged muscle groups and ligaments after healing, and the friction force during the movement of the hip joint and knee joint [13], etc., which will make the actual gait trajectory control effect in the process of movement less than ideal. Nowadays, PD control [14] and impedance control [15], etc., are commonly used in rope-driven lower limb rehabilitation robot, generally ignoring unknown interference in the process of motion and not making full use of periodic motion, which leaves more room for improvement in control accuracy. Therefore, the control scheme with the function of feedforward torque compensation and periodic repeated learning is more prominent.

Repetitive learning control is a control method with periodic repetitive motion, which has been applied in the field of exoskeleton-assisted robots and flexible space robots and shows excellent control performance [16,17]. At present, the main optimization direction of rope traction rehabilitation robots is the optimization of feedback parameters and disturbance estimation error compensation. However, in actual rehabilitation, different patients have different degrees of muscle damage, resulting in different external periodic disturbance, and fixed feedback compensation cannot be accurately applied to different patients. The control scheme with a repetitive learning function is more suitable for this kind of rehabilitation equipment.

### 3.2. Trajectory Planning and Dynamics Model of the Lower Limb Rehabilitation Robot

The movement mode and range of human lower limb joints are limited, with the range of hip joint flexion/extension angles being $-30°/120°$ and knee joint flexion/extension angles being $-120°/0°$ [18]. After determining the angle range, it is necessary to collect the angle changes during the motion process, so the infrared NOKOV capture system is used for dynamic capture experiments on the human body, as shown in Figure 5a,b.

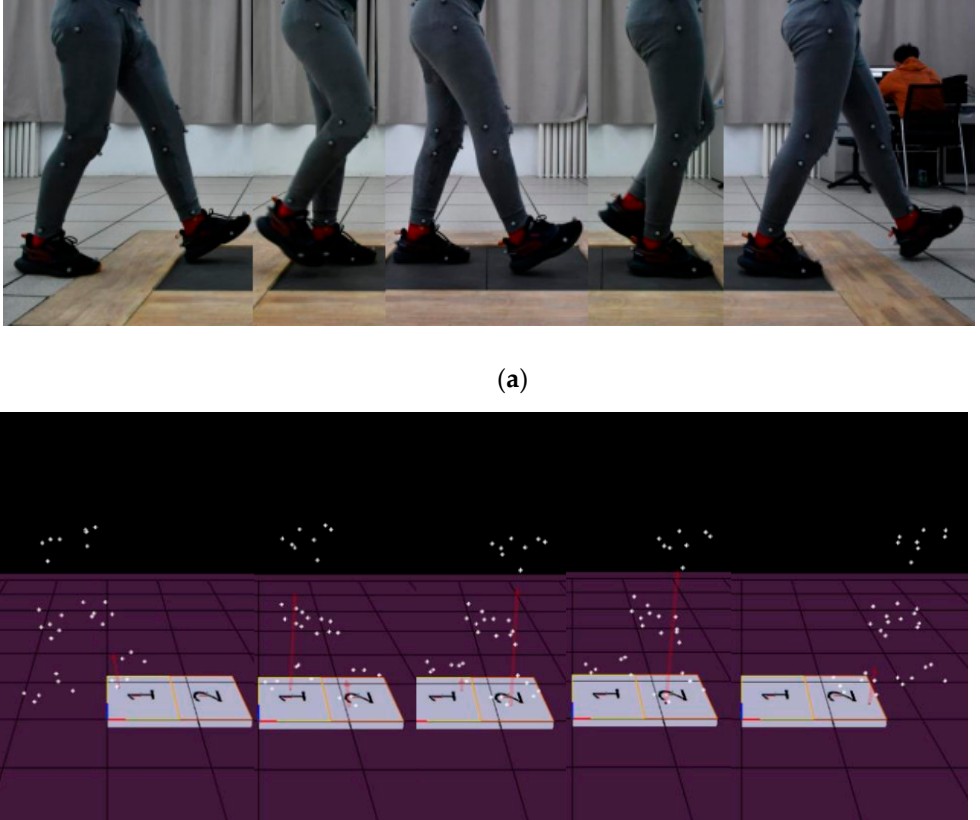

(**a**)

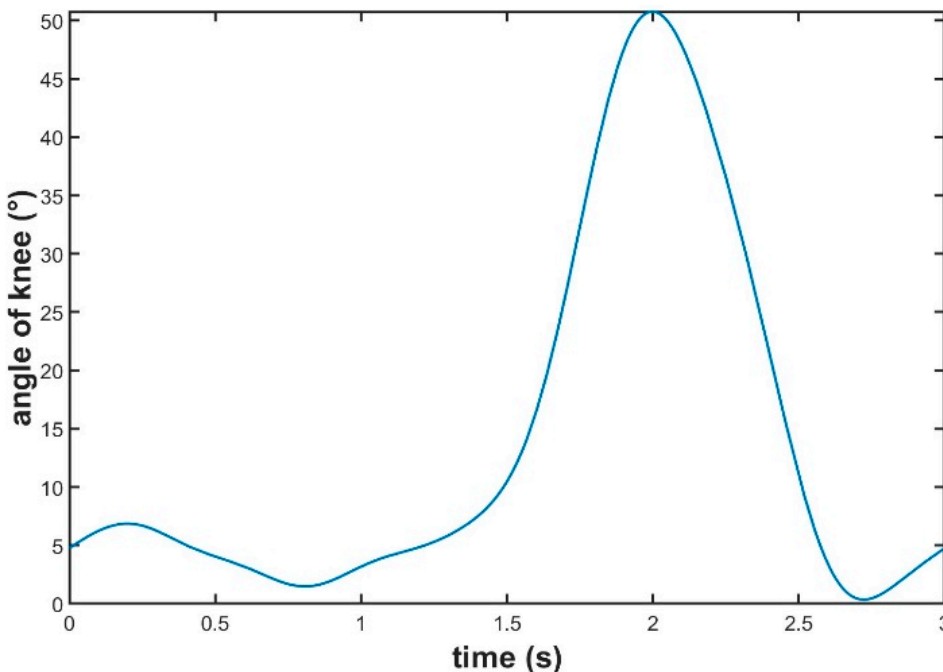

(**b**)

**Figure 5.** (**a**) Human dynamic capture experiment. (**b**) Human dynamic capture and collection.

The motion points of normal walking of the human body are obtained, and the motion trajectories are fitted by the eighth-order Fourier function. The trajectory of knee joint movement is shown in Figure 6.

**Figure 6.** Knee joint motion trajectory.

The knee joint motion function is:

$$\begin{aligned}
\theta_1 = 14.38 &- 7.089cos(wt) - 16.19sin(wt) - 8.119cos(2wt) + 9.122sin(2wt) + 4.434cos(3wt) \\
&+ 3.721sin(3wt) + 0.02693cos(4wt) - 0.842sin(4wt) + 0.3988cos(5wt) \\
&+ 0.7564sin(5wt) + 0.4781cos(6wt) - 0.146sin(6wt) - 0.182cos(7wt) \\
&+ 0.08417sin(7wt) - 0.008469cos(8wt) + 0.06557sin(8wt)
\end{aligned} \tag{1}$$

In the formula, $w$ = 2.094.

The motion trajectory of the hip joint is shown in Figure 7.

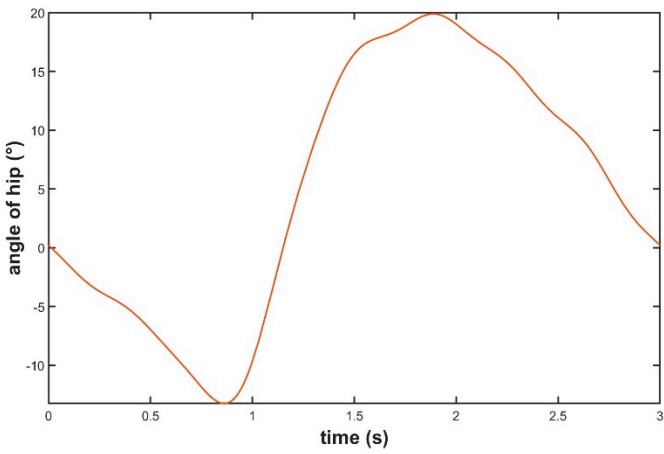

**Figure 7.** Hip joint motion trajectory.

The motion function of the hip joint is:

$$\begin{aligned}
\theta_2 = -5.081 &- 13.95cos(wt) - 4.602sin(wt) - 1.944cos(2wt) + 3.305sin(2wt) + 1.96cos(3wt) \\
&+ 0.5971sin(3wt) - 0.2532cos(4wt) - 1.001sin(4wt) - 0.3967cos(5wt) \\
&- 0.01045sin(5wt) + 0.1376cos(6wt) + 0.04932sin(6wt) - 0.08074cos(7wt) \\
&- 0.05228sin(7wt) - 0.2824cos(8wt) - 0.02406sin(8wt)
\end{aligned} \tag{2}$$

In the formula, $w$ = 2.094.

The dynamic model of lower limb rehabilitation robot can be expressed as:

$$\tau(t) = M(\theta)\ddot{\theta} + C\left(\theta, \dot{\theta}\right)\dot{\theta} + G(\theta) + E(\theta) + V\left(\dot{\theta}\right) + D(t) \tag{3}$$

In the formula, $M(\theta) \in R^{2 \times 2}$ represents the inertia matrix; $C\left(\theta, \dot{\theta}\right) \in R^{2 \times 2}$ represents the centripetal force and coriolis force matrix; $G(\theta) \in R^2$ represents the gravity vector; $E(\theta) \in R^2$ represents the elastic force vector of the joint; $V\left(\dot{\theta}\right) \in R^2$ represents the joint rotation when the viscous friction force vector is considered; $D(t) \in R^2$ represents the external periodic perturbative force vector; and $\tau(t) \in R^2$ represents the torque vector of the hip joint and knee joint [19,20].

### 3.3. Design of Repetitive Learning Control

The roped-driven lower limb rehabilitation robot studied in this paper makes the legs of patients move periodically and reciprocally by imitating the walking gait of normal human beings through the traction coupled with the rope. Therefore, we set the expected position trajectory of the active joint for periodic motion $\theta_e$, expected velocity trajectory $\dot{\theta}_e$, and the expected acceleration trajectory $\ddot{\theta}_e$.

We set the expectations trajectory tracking error $e_f$ and the desired speed trajectory tracking error $\dot{e}_f$, which are defined respectively as:

$$e_f = \theta - \theta_e, \ \dot{e}_f = \dot{\theta} - \dot{\theta}_e \tag{4}$$

In the formula, $\theta$ is the locus of the actual position and $\dot{\theta}$ is the actual speed trajectory.

To reduce the steady-state error in the control system, the introduction of the tracking error integral item $\int_0^t e_f dt$, define the position of reference trajectory $\theta_r$ as follows:

$$\theta_r = \theta_e - \Lambda \int_0^t e_f dt \tag{5}$$

In the formula, $\Lambda$ is a user-defined positive value. The reference velocity trajectory after derivative $\dot{\theta}_r$ is:

$$\dot{\theta}_r = \dot{\theta}_e - \Lambda e_f \tag{6}$$

For reference speed trajectory $\dot{\theta}_r$, use position tracking error $e_f$ correction expect for speed in order to ensure the convergence of the tracking error. When using the actual speed $\dot{\theta}$ to expect the speed $\dot{\theta}_e$ lag, the reference speed $\dot{q}_r$ increases.

The reference speed track tracking error $e_s$ is set and defined as [21]:

$$e_s = \dot{\theta} - \dot{\theta}_r = \dot{e}_f + \Lambda e_f \tag{7}$$

Using Formula (7) and the dynamics model of the lower limb rehabilitation robot, obtain the output $\tau(t)$ as:

$$\tau(t) = M(\theta)\left(\dot{e}_s + \ddot{\theta}_r\right) + C\left(\theta, \dot{\theta}\right)\left(e_s + \dot{\theta}_r\right) + G(\theta) + E(\theta) + V\left(\dot{\theta}\right) + D(t) \tag{8}$$

Setting $F\left(\theta, \dot{\theta}\right)$ as follows:

$$F\left(\theta, \dot{\theta}\right) = M(\theta)\ddot{\theta}_r + C\left(\theta, \dot{\theta}\right)\dot{\theta}_r + G(\theta) + E(\theta) + V\left(\dot{\theta}\right) \tag{9}$$

By substituting the above formula into Formula (8), its closed-loop dynamic model can be written as follows:

$$\tau(t) = M(\theta)\dot{e}_s + C\left(\theta, \dot{\theta}\right)e_s + F\left(\theta, \dot{\theta}\right) + D(t) \tag{10}$$

By (10), $M(\theta)$ and $C\left(\theta, \dot{\theta}\right)$ are a known quantity and $\dot{e}_s$ and $e_s$ are measurable values, as long as $F\left(\theta, \dot{\theta}\right)$ and $D(t)$ can obtain the real value output torque value. Although you cannot measure the $F\left(\theta, \dot{\theta}\right)$ and $D(t)$ of the real value, they can be measured by means of repetition in the process of tracking the periodic reciprocating motion of the lower limb with the constantly updated $F\left(\theta, \dot{\theta}\right)$ and $D(t)$ estimate, which after several cycles tends to be the real value, and this realizes the $F\left(\theta, \dot{\theta}\right)$ and $D(t)$ for the real value of learning.

About the calculation of $F\left(\theta, \dot{\theta}\right)$, because of the dynamic model of $M(\theta), C\left(\theta, \dot{\theta}\right)$, $G(\theta)$, $E(\theta)$, and $V\left(\dot{\theta}\right)$ related to the stiffness and damping of the lower limb joints, they are related to the impedance of the lower limb rehabilitation robot system dynamics, and the nature of which is as follows [22,23]:

**Theorem 1.** *Set $L = [l_1, l_2, l_3, l_4]^T \in R^4$ and $\Phi = \left[1, \|\dot{q}\|, \|\dot{q}\| \|\dot{q}_r\|, \|\ddot{q}_r\|\right]^T \in R^4$. There are a finite number of positive constants $l_w^* > 0 (w = 1, 2, 3, 4)$, for $\forall q \in R^2$ and $\forall \dot{q} \in R^2$, there exists $\|F(q, \dot{q})\| = \|M(q)\ddot{q}_r + C(q, \dot{q})\dot{q}_r + G(q) + E(q) + V(\dot{q})\| < L^{*T}\Phi = l_1^* + l_2^*\|\dot{q}\| + l_3^*\|\dot{q}\| \|\dot{q}_r\| + l_4^*\|\ddot{q}_r\|$.*

In Theorem 1, $\|\cdot\|$ represents a scalar value of a vector and $L^{*T} \in R^4$ represents the true value of the vector L for the stiffness and damping coefficients.

Therefore, the available $F\left(\theta, \dot{\theta}\right)$ as follows:

$$F\left(\theta, \dot{\theta}\right) = \hat{L}^T \Phi \tag{11}$$

In the formula, $\hat{L}^T$ is the $L^{*T}$ estimate.

Estimate $\hat{L}^T$ iterative learning based on the vector impedance update as follows [21,24]:

$$\dot{\hat{L}}(t) = -\psi(h(t)\hat{L}(t) - \|e_s\|\Phi \tag{12}$$

In the formula, $\dot{\hat{L}}(t)$ is the time function after learning iteration of the estimated value $\hat{L}(t)$; $\psi$ is a positive constant that affects the learning speed of L*;and h(t) is the custom time function, and to satisfy $h(t) > 0$, $\lim\limits_{t\to\infty} h(t) = 0$, $\int_0^\infty h(t)dt = a < \infty$.

As for the calculation of D(t), the estimated value can be calculated by means of a linear combination. Its function is to compensate the external periodic disturbance in the lower limb rehabilitation movement because the lower limb rehabilitation robot is driven by six ropes in one leg and can be converted into the torque output points of the knee and hip joints. Therefore, $D(t)$ has two components $D_i(t)_{(i=1,2)} \in R$, where each term in the linear combination is the product of a form function $\xi_j(t) \in R$, and a corresponding constant coefficient $k_j^{i*} \in R$, so $D_i(t)_{(i=1,2)}$ can be expressed as:

$$D_i(t) = \sum_{j=0}^{N-1} k_j^{i*}\xi_j(t) \tag{13}$$

In the formula, $N$ is the number of form functions; $k_j^{i*}$ is the exact value of the constant coefficient corresponding to the form function.

Due to the periodic movement of the lower limb rehabilitation robot, $D_i(t)$ also has periodicity and continuity, so the interval dense form function of the Stone–Weirstrass theorem can be used to approximate the exact value of $D_i(t)$, and the optional form function is defined as follows [25]:

**Definition 1.** *Set* $S(T)$ *to represent a periodic function subspace in continuous function space* $C[0, T]$, *which satisfies equal values of left and right endpoints, and consider a countable set of linearly independent* $\{\xi_j \in S(T)\}$ *in that:*

1. *The identity element can be expressed as a linear combination containing the finite term* $\xi_j$
2. *The span of* $\{\xi_j\}$ *is dense over* $S(T)$ , *that is, for any* $D_i(t) \in S(T)$ *and* $\zeta > 0$, *there are positive integers N and* $k_j^i$ *such that:*

$$\sup_{t\in[0,T]} |D_i(t) - \sum_{j=0}^{N-1} k_j^i\xi_j(t)| < \zeta \tag{14}$$

In this chapter, $N$ piecewise linear functions conforming to Definition 1 are selected as shape functions. They are defined as follows:

$$\beta_j(t) = \frac{N}{T}t - j \tag{15}$$

When $j = 0$, form function $\xi_j(t)$ in $t \in [0,T]$ can be expressed as:

$$\xi_j(t) = \begin{cases} 1 - \beta_j(t), & if\ 0 \le \beta_j(t) < 1 \\ 1 + \beta_j(t) - N, & if\ N-1 \le \beta_j(t) < N \\ 0, & else \end{cases} \tag{16}$$

When $j = 1, 2, \ldots, N-1$, the form function $\xi_j(t)$ of $t \in [0, T]$ can be expressed as:

$$\xi_j(t) = \begin{cases} 1 - \beta_j(t), & if\ \ 0 \le \beta_j(t) < 1 \\ 1 + \beta_j(t), & if\ -1 \le \beta_j(t) < 0 \\ 0, & else \end{cases} \tag{17}$$

In addition, the form function $\xi_j(t)$ is a time function that satisfies the period $T$.

After the shape function $\xi_j(t)$ is selected, the external periodic disturbance value $D_i(t)$ can be obtained by obtaining the approximate value of its corresponding coefficient $k_j^{i*}$. Since $D_i(t)$ is two perturbations, let $k_j^{i*} = \left[k_j^{1*}, k_j^{2*}\right]^T$, whose estimated value is

$\hat{k}_j^i = \left[\hat{k}_j^1, \hat{k}_j^2\right]^T$, and the periodic perturbation learning law of the designed estimated value $\hat{k}_j^i$ is:

$$\dot{\hat{k}}_j^i = -\Theta_j \frac{N}{2} \int_0^t e_s(t)\xi_j(t)d(t) \tag{18}$$

In the formula, $\dot{\hat{k}}_j$ is the iterative value of the estimated value $\hat{k}_j^i$; $\Theta_j$ is a positive constant that affects the learning speed of $\hat{k}_j$; $\frac{N}{2}$ is the reciprocal of the effective interval length (non-zero value) of the form function; and $e_s(t)\xi_j(t)$ is the correlation between velocity trajectory tracking error and $\xi_j(t)$, and their product value affects the value of constant coefficient $K_j$ of $\xi_j(t)$. The mean value of $e_s(t)\xi_j(t)$ correlation on time axis t is obtained through integration as the iteration value $\dot{\hat{k}}_j$.

Finally, according to Theorem 1 and Formulas (11) and (13), the repetitive learning controller for calculating the output torque of the joint can be put forward as:

$$\tau(t) = -K_p e_f - K_s e_s - sgn(e_s)\hat{L}^T + \sum_{j=0}^{N-1} \hat{k}_j \xi_j(t) \tag{19}$$

The rope-driven lower limb rehabilitation robot needs to achieve the same joint torque with the rope coupling torque. The space force sealing formula of the rope-driven rehabilitation robot is as follows:

$$\begin{bmatrix} U_1 & \cdots & U_i \\ {}_Q^O R \vec{Q A_1} \times U_1 & \cdots & {}_Q^O R \vec{Q A_i} \times U_i \end{bmatrix} \begin{bmatrix} f_1 \\ \vdots \\ f_i \end{bmatrix} = \begin{bmatrix} F \\ \tau(t) \end{bmatrix} \tag{20}$$

In the formula, $U_i$ is the unit vector of the i rope, ${}_Q^O R$ is the rotation matrix, and $\vec{Q A_i}$ is the position vector of the traction point $A_i$ on the local coordinate system Q. Set the force Jacobian matrix $A' = \begin{bmatrix} U_1 & \cdots & U_i \\ {}_Q^O R \vec{Q A_1} \times U_1 & \cdots & {}_Q^O R \vec{Q A_i} \times U_i \end{bmatrix}$ in the horizontal plane, the tension of each rope $T = \begin{bmatrix} f_1 \\ \vdots \\ f_i \end{bmatrix}$, and the resultant force of a single limb on the horizontal plane $W = \begin{bmatrix} F \\ \tau(t) \end{bmatrix}$. Therefore, Formula (20) is transformed into:

$$A'T = W \tag{21}$$

In lower limb rehabilitation training, patients carry out rehabilitation training with in situ reciprocating exercise. The reaction force generated by the bottom of the thigh under the translational displacement of the lower leg is regarded as the fifth traction rope, and the lower leg and thigh are only subjected to torque, namely, $W = \begin{bmatrix} 0 \\ \tau(t) \end{bmatrix}$. Given the required torque W and force Jacobian matrix $A'$, the tension T of each rope can be calculated:

$$T = A'^+ W \tag{22}$$

In the formula, $A'^+$ is the generalized inverse of matrix $A'$.

The structure of the repetitive learning control system in passive rehabilitation mode is shown in Figure 8 as follows:

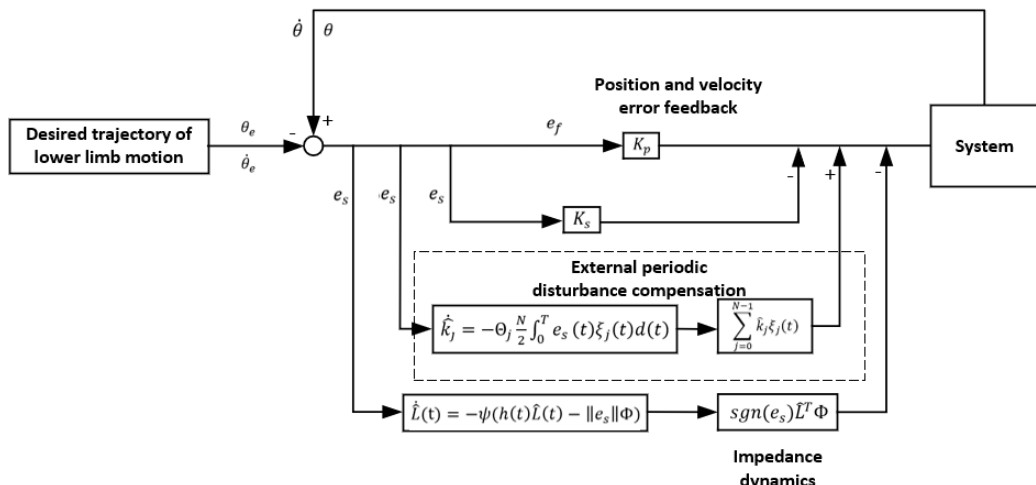

**Figure 8.** Structure of the passive rehabilitation repetitive learning control system.

In this chapter, the passive rehabilitation link of the rope-driven lower limb rehabilitation robot is optimized by the repeated learning control method. The control scheme consists of three control rings:

1. Formula (12) was used to learn the accurate value $L^{*T}$ of the impedance dynamic parameters of the lower extremity, and $F\left(\theta, \dot{\theta}\right)$ was estimated according to Theorem 1.

2. Based on the periodic external disturbance characteristics of rehabilitation training and the Stone–Weirstrass theorem, formulas (13) and (18) are established to solve the external periodic disturbance $D_i(t)$ and the external disturbance feedforward compensation is carried out.

3. Use $-K_p e_f$ to achieve the expected position error feedback and $-K_s e_s$ to achieve the reference speed error feedback.

Finally, the repetitive learning control is realized, which makes the lower limb rehabilitation robot achieve higher control precision.

## 4. System Stability Proof

The stability analysis of the control system of the rope-driven lower limb rehabilitation robot is proven by the stability theory of Lyapunov [26].

Theorem 1 considers nonlinear systems:

$$\dot{x}(t) = f(x(t), t) f(0, t) \equiv 0$$

If there exists a scalar function $V(x, t)$ with continuous first-order partial derivatives, the following conditions are satisfied:

(1) $V(x, t)$ is positive definite;

(2) The derivative of $V(x, t)$ with respect to time $\frac{dV(x,t)}{dt}$ is negative definite; then, the system is asymptotically stable everywhere at the origin.

To set the initial state of $\theta(0)$, $\dot{\theta}(0)$, $\hat{L}(0)$, $\hat{k}_j(0)$, if desired location error $e_f$, speed trajectory error $e_s$, estimate error $\widetilde{L}$ of the impedance dynamic coefficient $L^{*T}$ and estimate error $\widetilde{k}$ periodic disturbance coefficient $k^*$ all converge to zero over time, there is system stability. According to the closed dynamics model and two kinds of learning rates, the Lyapunov function is selected:

$$V(\eta, t) = V_j + V_s + V_l + V_k \tag{23}$$

According to Formula (6) setting:

$$V_j = \frac{1}{2} e_j^T K_p e_j \tag{24}$$

To solve its derivative is:

$$\dot{V}_j = -e_j^T K_p \Lambda e_j + e_j^T K_p e_s \tag{25}$$

According to Formulas (10) and (19) setting:

$$V_s = \frac{1}{2} e_s^T M(\theta) e_s \tag{26}$$

Its derivative is:

$$
\begin{aligned}
\dot{V}_s &= \frac{1}{2} e_s^T M(\dot{\theta}) e_s + e_s^T \left[ \tau(t) - C\left(\theta, \dot{\theta}\right) - F\left(\theta, \dot{\theta}\right) + D(t) \right] \\
&= \frac{1}{2} e_s^T \left( M(\dot{\theta}) - 2C\left(\theta, \dot{\theta}\right) \right) e_s + e_s^T \left[ -K_p e_f - K_s e_s - sgn(e_s) \hat{L}^T \Phi + \sum_{j=0}^{N-1} \hat{k}_j \xi_j(t) - \quad F\left(\theta, \dot{\theta}\right) + D(t) \right] \\
&\leq e_s^T \left[ -K_p e_f - K_s e_s - \|e_s\| \hat{L}^T \Phi \ + \|e_s\| L^{*T} \Phi + \sum_{j=0}^{N-1} \hat{k}_j \xi_j(t) \ - \sum_{j=0}^{N-1} k_j^* \xi_j(t) \right] \\
&\leq e_s^T \left[ -K_p e_f - K_s e_s - \|e_s\| \widetilde{L}^T \Phi + \sum_{j=0}^{N-1} \widetilde{k}_j \xi_j(t) \right]
\end{aligned}
\tag{27}
$$

In the formula, $M(\dot{\theta}) - 2C\left(\theta, \dot{\theta}\right)$ is the skew symmetric matrix according to the dynamics theorem and is negative definite.

According to Formula (12) setting:

$$V_l = \frac{1}{2} \int_{t-T}^{t} \widetilde{L}^T(x) \psi^{-1} \widetilde{L}(x) dx \tag{28}$$

Its derivative is:

$$
\begin{aligned}
\dot{V}_l &= \frac{1}{2} \left[ \widetilde{L}^T(t) \psi^{-1} \widetilde{L}(t) - \widetilde{L}^T(t-T) \psi^{-1} \widetilde{L}(t-T) \right] \\
&= \frac{1}{2\psi} \left( \widetilde{L}(t) - \widetilde{L}(t-T) \right)^T \left( \widetilde{L}(t) - \widetilde{L}(t-T) \right) = \frac{1}{2\psi} \dot{\widetilde{L}}^T \left( 2\widetilde{L}(t) - \dot{\widetilde{L}} \right) \\
&= \|e_s\| \widetilde{L}^T \Phi - \frac{1}{2\psi} \dot{\widetilde{L}}^T \dot{\widetilde{L}} + h(t) \hat{L}^T(t) \ \left( L^* - \hat{L}(t) \right) \\
&\leq \|e_s\| \widetilde{L}^T \Phi - \frac{1}{2\psi} \dot{\widetilde{L}}^T \dot{\widetilde{L}} h(t) + \left( \frac{1}{2} L^{*T} L^* \right)
\end{aligned}
\tag{29}
$$

According to Formula (18) setting:

$$V_k = \sum_{j=0}^{N-1} \widetilde{k}_j^T \Theta_j^{-1} \widetilde{k}_j \tag{30}$$

Its derivative is:

$$\dot{V}_k = N \sum_{j=0}^{N-1} \widetilde{k}_j^T \widetilde{k}_j \ e_s(t) \xi_j(t) \tag{31}$$

Since $V(\eta, t)$ is positive definite, by satisfying the first term of Theorem 1, the system can be proven stable as long as its derivative is negative definite, and the derivative of $V(\eta, t)$ is:

$$
\begin{aligned}
\dot{V(\eta, t)} &= \dot{V}_j + \dot{V}_s + \dot{V}_l + \dot{V}_k \\
&\leq -e_j^T K_p \Lambda e_j + e_j^T K_p e_s + e_s^T \left[ -K_p e_f - K_s e_s - \|e_s\| \widetilde{L}^T \Phi + \sum_{j=0}^{N-1} \widetilde{k}_j \xi_j(t) \right] + \|e_s\| \widetilde{L}^T \Phi - \frac{1}{2\psi} \dot{\widetilde{L}}^T \dot{\widetilde{L}} + h(t) \left( \frac{1}{2} L^{*T} L^* \right) + N \sum_{j=0}^{N-1} \widetilde{k}_j^T \widetilde{k}_j \ e_s(t) \xi_j(t) \\
&\leq -e_j^T K_p \Lambda e_j - e_s^T K_s e_s + \frac{1}{2} h(t) L^{*T} L^*
\end{aligned}
\tag{32}
$$

Since Formula (12) is set to satisfy $\lim\limits_{t\to\infty} h(t) = 0$ and $\int_0^\infty h(t) dt = a < \infty$, when $t \to \infty$:

$$\lim_{t\to\infty} \frac{1}{2} h(t) L^{*T} L^* = 0 \tag{33}$$

And because $K_p$, $\Lambda$, $e_s$ for the positive definite matrix can determine $\dot{V(\eta, t)}$ as negative.

In conclusion, $V(\eta, t)$ is positive definite and $\frac{dV(\eta, t)}{dt}$ is negative definite, which satisfies the Lyapunov stability condition and proves that the control system is asymptotically stable.

## 5. Simulation Verification and Analysis

In order to verify the effectiveness and stability of the above repetitive learning system, this paper will use repetitive learning control and PD control to compare the performance of the motion position and speed of the joints, respectively, so as to verify that the repetitive learning control has a more stable and accurate control effect in the face of external periodic disturbance.

During the experimental process, the normal walking trajectory of the human body was obtained through dynamic capture experiments and used as the expected function at the input end. The iterative learning algorithm and impedance dynamics formula in Section 3.3 were used to establish periodic disturbance modules and impedance modules to achieve periodic error compensation and impedance approximation estimation, respectively. Periodic step disturbance signals were added to simulate unknown external periodic disturbances. Then, the dynamic structure model in Figure 9 was established through Simulink. After constantly adjusting the gain values and learning speed coefficients, the desired control effect was finally obtained.

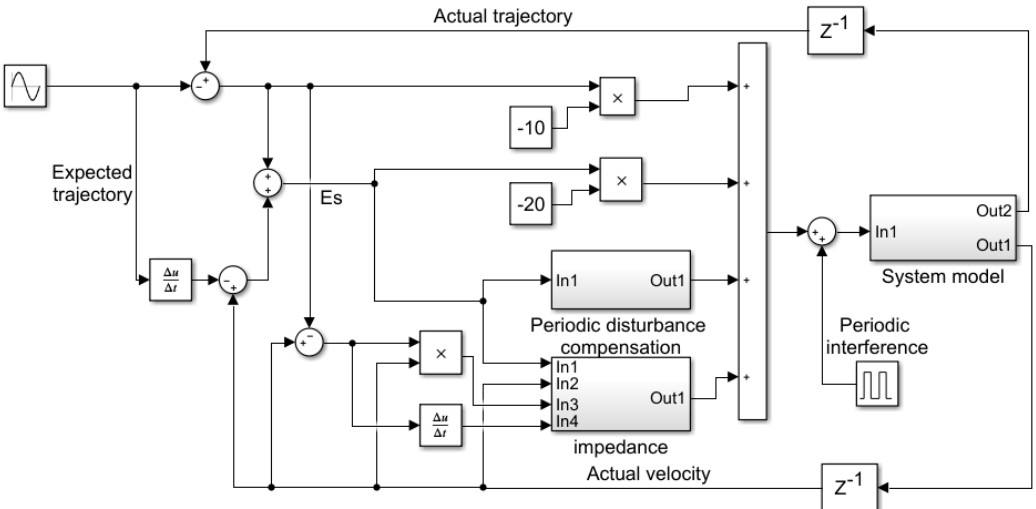

**Figure 9.** Repetitive learning control simulation model.

The simulation experiment parameters are shown in Table 2:

**Table 2.** Parameter values of the repetitive learning control structure model.

| Parameter | Numerical Value |
|:---:|:---:|
| $\Lambda$ | 1 |
| $K_p$ | 10 |
| $K_s$ | 20 |
| N | 20 |
| $\Psi$ | $2 \times 10^{-7}$ |
| $h(t)$ | $1/(1+t)^2$ |
| $\Theta_i$ | 5 |

Through simulation experiments, the tracking error results of repetitive learning control and PD control can be obtained, as shown in Figures 10 and 11. Figure 10 shows the different position tracking effects of the same joint under repetitive learning control and PD control. The blue dashed line represents the error fluctuation through PD control, and the tracking error amplitude has almost no change with the periodic fluctuation of the input signal and external disturbance signal. The red solid line represents the periodic repetitive

learning control system, and the tracking error continues to decrease and stabilize with the continuous repetition of the gait cycle, which realized the early compensation function for external periodic disturbances. Figure 11 shows the comparison of speed tracking errors achieved by repetitive learning control and PD control. It can be seen that regardless of how long it has been since the motion cycle has passed, the speed error under PD control always fluctuates steadily with the cycle. However, repetitive learning control significantly reduces the range of speed fluctuation values after a few motion cycles, which can prove that repetitive learning control has better performance in speed tracking.

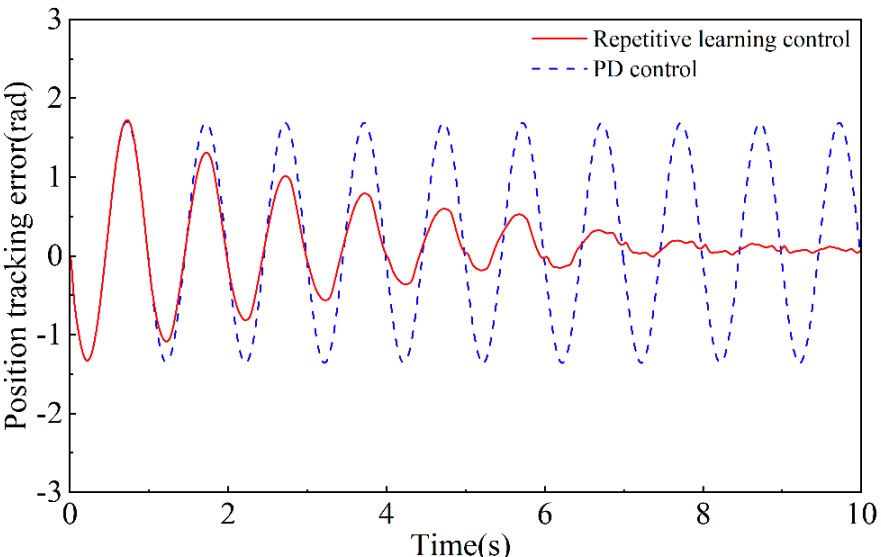

**Figure 10.** Comparison of position tracking errors.

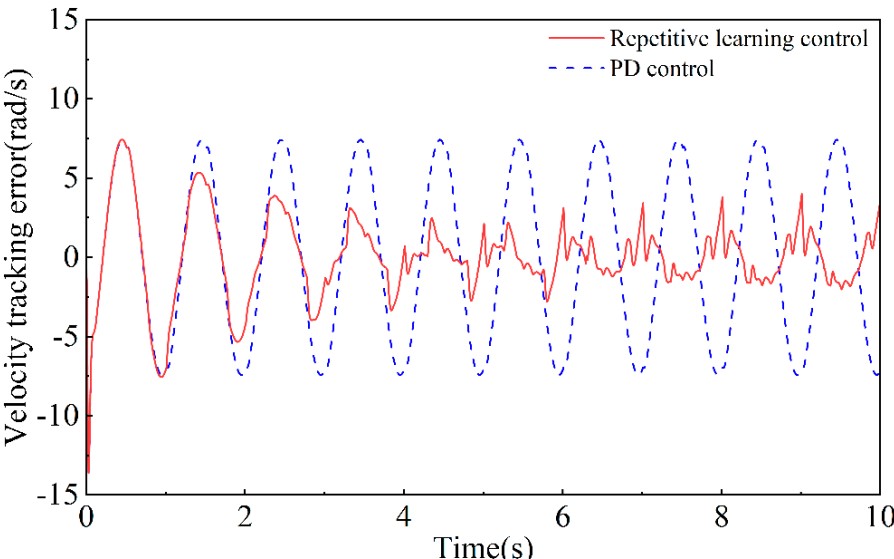

**Figure 11.** Comparison of speed tracking errors.

## 6. Conclusions

In this paper, a periodic disturbance compensation control method was proposed to solve the approximations of the external periodic disturbance term and the unknown dynamic term in order to solve the problem of the external periodic disturbance term encountered by the rope-driven lower limb rehabilitation robot during its operation, and the relevant parameters were constantly learned and updated during each gait cycle. The control algorithm was verified by simulation and compared with the PD control results.

It can be concluded that the periodic repetitive learning control can continuously shorten the tracking error in the process of motion and reach the ideal value in a short time, which proves the accuracy and effectiveness of the control method. The experimental results show that the periodic disturbance compensation control can make the lower limb rehabilitation robot obtain a more accurate adaptive control effect in the face of patients with different physical indicators and effectively solve the problem of body impedance caused by differen patients.

**Author Contributions:** Conceptualization, Z.W. and M.L.; methodology, M.L.; software, M.L.; validation, M.L. and X.Z.; formal analysis, M.L.; resources, Z.W.; data curation, X.Z.; writing—original draft preparation, M.L.; writing—review and editing, M.L.; supervision, Z.W. All authors have read and agreed to the published version of the manuscript.

**Funding:** This research was funded by National Natural Science Foundation of China (No. 51505124), Science and Technology Project of Hebei Education Department (ZD2020151), and Tangshan Science and Technology Innovation Team Training Plan Project (21130208D).

**Data Availability Statement:** No new data were created or analyzed in this study. Data sharing is not applicable to this article.

**Conflicts of Interest:** The authors declare no conflict of interest. The funders had no role in the design of the study; in the collection, analysis, or interpretation of the data; in the writing of the manuscript; or in the decision to publish the results.

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
