# Peer review of "Periodic Disturbance Compensation Control of a Rope-Driven Lower Limb Rehabilitation Robot"

_actuators, doi:10.3390/act12070284_

Round 1
Reviewer 1 Report
The article designs a rope driven lower limb rehabilitation robot and uses a periodic compensation control to effectively solve the problem of periodic disturbances generated during passive control of the robot. The topic is interesting and the paper may have some merits. Here are some of my comments.
1. The repetitive learning control method is not only effective for periodic disturbances in lower limb rehabilitation robots, but also has great reference value for other robots that are subject to periodic disturbances. Please provide some discussion.
2. The presentation needs to be improved. For example, Fig. 5a and Fig. 5b share the same caption. Equations are not cited corrected, i.e. you can just say "By (10)", instead of "By fomula (10)". In (32), there is typo of derivative of V. There is a value without paramter name in Table 2.
3. Authors need to disucss the obtained results after the proof. For example, the purpose of V_j,V_s,V_l,V_k, the advantages and disadvantages.
4. Reference needs to be updated with some recent results on this topic.
Authors need to improve the quality of the English.
Reviewer 2 Report
Paper presents a control technique for a rope driven lower limb exoskeleton robot, the study is interesting however,
Introduction
All statistical facts must be cited, for example in the 1st paragraph when the WHO data is mentioned.
A more exhaustive state of the art is needed
Control design
It is not clear neither which motion of the knee and hip is considered (flexion/extension, abduction/adduction, int/ext rotation), nor why the angle function is determined.
Simulation
It is not clear the simulation process, neither how the simulation data was obtained nor the gait parameters used
Gait data from subjects needs to be add and evaluate comparing with other studies.
A discussion of the results must be added, the conclusions needs to be improved.
English is fine.
Round 2
Reviewer 2 Report
The following recommendation must be addressed before accepting
Control design
It is not clear neither which motion of the knee and hip is considered (flexion/extension, abduction/adduction, int/ext rotation), nor why the angle function is determined.
Simulation
It is not clear the simulation process, neither how the simulation data was obtained nor the gait parameters used
Gait data from subjects needs to be add and evaluate comparing with other studies.
You can find relevant information for support the experiments in:
https://www.mdpi.com/1424-8220/20/3/789
A discussion of the results must be added, the conclusions needs to be improved.
English is fine
Author Response
Thank you very much for the valuable revision suggestions provided by the reviewers. After reading the Conceptual Development of a Lower Limb
The Exoskeleton Robot Driven by an On Board Musculoskeletal Simulator article in this related direction was greatly inspired and learned a lot of cutting-edge knowledge. Through the detailed data collection information in its article, it once again improved my article.
- Regarding the insufficient experimental data, some references[18] have been added.
- In terms of control design, the joint motion mode has been added in lines 110-115, and the angle function is used as the expected trajectory function.
- Regarding the simulation, the simulation process has been uploaded as an attachment.
Finally, thank you once again for the help and guidance you provided for my manuscript.
Round 3
Reviewer 2 Report
All recommedations has beed addressed.